# Synthesis of New Indanyl Nucleoside Analogues and their Biological Evaluation on Hepatitis C Virus (HCV) Replicon

**DOI:** 10.3390/molecules24050990

**Published:** 2019-03-11

**Authors:** Matías E. Gómez, Emiliano A. Gentile, M. Florencia Martini, María L. Cuestas, Verónica L. Mathet, Graciela Y. Moltrasio, Albertina G. Moglioni

**Affiliations:** 1Cátedra de Química Medicinal, Facultad de Farmacia y Bioquímica, Universidad de Buenos Aires, Ciudad Autónoma de Buenos Aires C1113AAD, Argentina; matgmz@gmail.com (M.E.G.); flormartini1@gmail.com (M.F.M.); 2Instituto de Investigaciones en Microbiología y Parasitología Médica, CONICET-Universidad de Buenos Aires, Ciudad Autónoma de Buenos Aires C1121ABF, Argentina; emilianogentile@gmail.com (E.A.G.); marilu_cuestas@yahoo.com.ar (M.L.C.); vmathet@gmail.com (V.L.M.); 3Instituto de la Química y Metabolismo del Fármaco, CONICET-Universidad de Buenos Aires, Ciudad Autónoma de Buenos Aires C1113AAD, Argentina; 4Cátedra de Química Orgánica II, Facultad de Farmacia y Bioquímica, Universidad de Buenos Aires, Ciudad Autónoma de Buenos Aires C1113AAD, Argentina; gmoltra@gmail.com

**Keywords:** nucleoside analogues, *cis*-2-amino-1-indanol, HCV replicon, molecular modeling

## Abstract

Here, we report a convenient synthetic procedure for the preparation of four novel indanyl carbanucleoside derivatives in the racemic form. The action of these compounds against hepatitis C virus was evaluated in vitro using the replicon cell line, Huh7.5 SG. Contrary to our expectations, all these compounds did not inhibit, but rather promoted HCV genotype 1b (HCVg1b) replication. Similar effects have been reported for morphine in the replicon cell lines, Huh7 and Huh8. Several biological experiments and computational studies were performed to elucidate the effect of these compounds on HCVg1b replication. Based on all the experiments performed, we propose that the increase in HCVg1b replication could be mediated, at least in part, by a similar mechanism to that of morphine on the enhancement of this replication. The presence of opioid receptors in Huh7.5 SG cells was indirectly determined for the first time in this work.

## 1. Introduction

Interest in the design and synthesis of modified nucleosides has increased steadily over the past few decades [1,2,3,4,5]. The introduction of structural diversity into the nucleoside scaffold for use as potential chemotherapeutic agents has long been considered an important approach to drug design. In this sense, the substitution of the furan ring with a cyclopentane ring, and the subsequent replacement of both at positions 2’ and 3’ by an aromatic or heteroaromatic cycle (Figure 1: I and II, respectively), lead to differences in the physicochemical properties of the synthesized carbanucleosides, which may influence the biological activity due to higher lipophilicity, conformational rigidity, and steric hindrance than their natural nucleosides [6].

In this regard, we have previously designed and synthesized numerous carbanucleosides derived from the four commercially available stereoisomers of 1-amino-2-indanol [7] using the short, efficient, and linear synthetic route proposed by Shealy and Clayton [8], which showed interesting results as antiviral agents [9].

Persistent infection with hepatitis C virus (HCV) is associated with chronic hepatitis (CHC), cirrhosis, and the development of hepatocellular carcinoma (HCC). The current therapeutic approach is based on the combined use of direct-acting antiviral agents targeting the viral polymerase, the viral protease, and NS5A, with or without ribavirin [10,11,12,13,14,15,16]. Therefore, the study of direct-acting antivirals against hepatitis C continues to evolve at a rapid pace.

Taking into account our previous results [7,9], the literature available [17,18,19,20] regarding the activity observed for other indanyl nucleosides against different viruses, the fact that carbanucleosides derived from 2-amino-1-indanol have not been previously synthesized and evaluated as antiviral agents, and the persistent need to find effective new drugs for the treatment of hepatitis C, we decided to synthesize this type of derivative that contains triazole, purine, or azapurine moiety in their structures. Once synthesized, we evaluated their anti-HCV activity in vitro, by using the replicon cell line, Huh7.5 SG, to find new agents against this infection. The results obtained fell short of our expectations, as in all the cases, the synthesized compounds increased HCV replication. In this context and considering that the molecular analysis of these nucleoside analogues is relevant for explaining the effect observed, the aim of this work was expanded to search for the atomic grouping present in these derivatives, which are responsible for this unexpected effect. 

## 2. Results and Discussion

### 2.1. Chemistry

Based on our previous experience with the synthesis of indanyl nucleosides [7,9] and considering that 2-amino-1-indanol is not commercially available in any possible stereoisomeric form, we attempted a synthetic method that would lead to 2-amino-1-indanol with a defined *cis* stereochemistry and good yield. For this purpose, commercially available indene was treated with *N*-bromosuccinimide (NBS) to give (+/−) *trans*-bromohydrin **1** [21,22]. Thus, compound **1** was treated with ammonium hydroxide to yield a mixture of products that could not be identified; however, the treatment of compound **1** with sodium azide led to the isolation of the corresponding racemic azide 2 [23] with a very good yield (Scheme 1).

The complexity of the observed signals for the aliphatic H atoms of compound **2**, despite corresponding to the chemical displacements already described [23,24], did not allow the expected *cis* stereochemistry to be ensured, which could be defined through the bidimensional NOESY (Nuclear Overhauser SpectroscopY) spectra (Appendix A), where the nuclear Overhauser effect (nOe) between H1 and H2 of the indanyl ring was observed, indicating that both are on the same face of the molecule and confirming the *cis* stereochemistry. Then, compound **2** was reduced by catalytic hydrogenation to the desired (+/−)-*cis* -2-amino-1-indanol **3** [24]. Indanyl carbonucleosides were prepared using the classical approach to carbocyclic nucleosides proposed by Shealy and Clayton [8,20]. In this way, the condensation of compound **3** with 5-amino-4,6-dichloropyrimidine in refluxing *n*-butanol containing triethylamine afforded compound **4**. To obtain the imidazole ring of the purinyl analogues, compound **4** was treated with triethyl orthoformate, using a catalytic amount of hydrochloric acid, to render compound **5** in a 73% yield. The *cis* configuration between the hydroxyl group in C1, of the indane ring, and the 6-chloropurine residue of compound **5** was determined using HSQC (Heteronuclear Single Quantum Coherence) and NOESY correlation spectra (Appendix A). A triazole ring was formed from compound **4** by intramolecular reaction of the diazonium salt of the primary amino group with sodium nitrite, in an acidic medium, to render a highly unstable compound **6** (not isolated), which was converted into the 8-aza purine derivative **7** by heating. Compound **6** treated with ammonium hydroxide afforded compound **8**. Finally, triazole **9** was synthesized by treating compound **2** with propargyl alcohol through the Huisgen dipolar cycloaddition of azides and alkynes, a powerful class of the concerted click reaction, which were widely applied [25]. This reaction was catalyzed by Cu (II) in the presence of sodium ascorbate. All the compounds obtained were characterized by spectroscopic methods.

### 2.2. Biological Results

#### 2.2.1. Cytotoxicity of Indanyl Derivatives

Huh7.5 SG cells were incubated at different concentrations (1.56 to 100.00 µM) of the four newly synthesized carbanucleoside analogues (**5**, **7**–**9**) and cytotoxicity was estimated using the MTS/PMS ((3-(4,5-dimethylthiazol-2-yl)-5-(3-carboxymethoxyphenyl)-2-(4-sulfophenyl)-2H-tetrazolium inner salt)/phenazine methosulfate) assay. Results showed that the exposure of Huh7.5 SG cells to 25.00 µM of compound **5** and to 100.00 µM of nucleoside analogues **7**–**9**, for 96 h did not significantly decrease cell viability when compared to their corresponding controls. In contrast, 50.00 and 100.00 µM of compound **5** caused a significant cell viability loss in the HCV replicon cell line (Figure 2).

#### 2.2.2. HCVg1b Replication Assay

Compounds **5**, **7**–**9** were assayed at the highest concentration in which no cytotoxicity was observed: 25.00 µM (for compound **5**) and 100.00 µM (for compounds **7**–**9**). Surprisingly, as shown in Figure 3, there was a significant increase in the RNA levels of HCV in those Huh7.5 SG cells treated with all the derivatives: Compound **5** and **7** (more than 2-fold); compound **8** (approximately 5-fold); and compound **9** (nearly 6-fold), after 72 h post-incubation.

Since none of the tested compounds yielded the expected antiviral activity, we found it was not worthwhile to assay the activity as a function of the concentration of these compounds. 

Given these results, our investigations were redirected to the search of the structural motif that was responsible for the biological activity observed. Thus, considering the fact that the *cis*-2-amino-1-indanol moiety (**3**) is common to all the nucleoside analogues tested, and that the 6-chloropurine residue, only contained in compound **5**, is commercially available, a new experiment was conducted by testing the action of compounds **3**, **5**, and 6-chloropurine on viral replication. Although it was observed that compound **3** promotes an increase in viral load replication, its effect was lower than in all the nucleoside analogues assayed. Moreover, 6-chloropurine promoted viral replication, but in a lower proportion than *cis*-2-amino-1-indanol (Figure 4).

Similar increases in the viral loads to those observed in this work were previously reported in the literature by Li et al. [26,27]. They reported that Huh7.0, Huh.8, and FCA-1 cells express the μ-opioid receptor and it was proposed that the increase in HCV replication in morphine addicts is due to the interaction between this drug and the abovementioned receptor. Since Li et al. [26] showed that the effect on viral replication is time-dependent, we also decided to measure the HCV RNA at 48 h post-incubation for compounds **3**, **5**, and 6-chloropurine (Figure 4). Just as these authors had previously reported, the viral loads in our cell system were also higher at 48 h post-incubation, with respect to our observations at 72 h.

#### 2.2.3. Indanyl Nucleoside Activity Related to the Opioid Receptor

As was mentioned in 2.2.2, some abuse drugs, such as morphine, have been described as compounds that enhance HCV replicon expression [26,27]. In addition, naltrexone and naloxone, opioid receptor antagonists, abrogate the enhancing effect of that abuse drug on HCV replicon expression [26,27]. To elucidate whether the Huh7.5 SG replicon system responds to morphine in a similar way as the mentioned cells, an assay with morphine at 1.00 µM was conducted at 48 h and 72 h (Figure 5). The morphine treatment increased viral replication, and the effect was greater at 48 h. However, pretreating the cell culture with naloxone (0.01 µM) for 30 min inhibited the ability of morphine to exert its effect. These results suggest the presence of an opioid receptor in this subgenomic HCV replicon system.

Based on these results and considering that morphine-related compounds [28], such as phenylpiperidines (prodines), share the pharmacophore group that could be present in the assayed indanyl nucleosides (Figure 6), Huh7.5 SG cells were pretreated for 30 min with naloxone and then compound **5** was added to elucidate whether this compound can upregulate HCVg1b replication in a similar manner to morphine.

A drop in HCVg1b replication was observed when the Huh7.5 SG cells were preincubated with naloxone for 30 min and then compound **5** was added (Figure 7).

Taken together, these results led us to conclude that the increase in viral replication in the presence of nucleoside analog **5** could be mediated, at least partly, by the opioid receptor in the Huh7.5 SG cells, since a decrease in viral replication was observed when preincubation with naloxone was performed. A similar mechanism could be responsible for the action of compounds **7**–**9**.

To further evaluate if compound **5** exerts its function through the opioid receptor, a combined treatment with morphine was assayed. Unexpectedly, the effect of the combined treatment displayed a lower HCVg1b replication level compared to compound **5** alone, but higher than that observed when cells were treated only with morphine. A possible explanation of this phenomenon could be that although morphine seems to have a higher affinity for the opioid receptor than compound **5**, the latter compound induces higher levels of HCVg1b replication (Figure 8), which reaffirms its effect, as previously mentioned.

An assay demonstrating the activity of a racemic mixture does not assure or rule out the activity of both enantiomers. Nevertheless, given the unexpected activity of compound **5** on HCVg1b replication, regardless of whether either of the two enantiomers or both were responsible for the observed activity, we attempted to establish a possible structural similarity between each enantiomer of compound **5** and morphine. In this way, molecular modeling studies for the two enantiomers were carried out. Both enantiomers of compound **5** and morphine were structurally optimized according to the procedure described in the Materials and Methods section. Each one of the 10 conformers obtained was superimposed on the structure of morphine, considering the atoms involved in the pharmacophore group (Figure 6). The best fitting structures of both enantiomers in compound 5 and the morphine structure are shown in Figure 9.

Since both enantiomers of compound **5** had a low RMSD value (0.274 for 1*S*,2*R* and 0.275 for 1*R*,2*S*), we were able to verify that their overlap with the morphine pharmacophore was adequate. This fact could lead us to the preliminary conclusion that both enantiomers are capable of exerting the observed effect.

## 3. Materials and Methods 

### 3.1. Chemical Experimental Section

Melting points were determined on a Thomas Hoover apparatus (Arthur H. Thomas Company, Philadelphia, PA, USA). ^1^H and ^13^C NMR spectra were recorded on Bruker 600 or a Bruker 300 spectrometers (Bruker Biospin, Rheinstetten, Germany), and reported in parts per million (δ) relative to tetramethylsilane (TMS: δ 0.00 ppm). Data are reported as follows: Chemical shift, multiplicity (s = singlet, d = doublet, t = triplet, m = multiplet, dd = doublet of doublet, ddd = doublet of doublets of doublets, bs = broad singlet), coupling constants (*J*, Hz), and integration. High resolution mass spectra (HRMS) were recorded on a micrOTOF-Q mass spectrophotometer (BRUKER DALTONIK, Bremen, Germany). Preparative thin layer chromatography (p-TLC) and thin layer chromatography analyses (TLC) were performed on Kieselgel 60 F254 (Merck KGaA, Darmstadt, Germany) plates.

#### 3.1.1. (+/−)-*trans*-2-bromo-1-indanol (**1**)

A solution of indene (0.4 mL, 3.4 mmol), NBS (0.67 g, 3.8 mmol), water (3.5 mL), and tetrahydrofurane (THF) (3.5 mL) was stirred for 12 h at room temperature. Product **1** was obtained as a white solid (0.68 g, 94%), which was filtered and crystallized from ethanol; m.p.: 127–129 °C; m.p. lit.: 132–133 °C [21,22]. Compound **1** was used in the next step without any further purification. ^1^H NMR (300 MHz, CDCl_3_) δ 2.45 (d, *J* = 5.6 Hz, 1H, OH); 3.20 (dd, *J* = 7.4 Hz, *J* = 16.2 Hz, 1H, CHH); 3.58 (dd, *J* = 7.2 Hz, *J* = 16.2 Hz, 1H, CHH); 4.28–4.38 (m, 1H, CHBr); 5.32 (t, *J* = 5.5 Hz, 1H, CHOH); 7.22–7.33 (m, 3H, ArH); 7.40–748 (m, 1H, ArH-7) ppm. ^13^C NMR (75.4 MHz, CDCl_3_) δ 41.1 (CH_2_), 55.1 (CHBr), 84.0 (CHOH), 124.8, 125.2, 128.3, 129.7, 140.4, and 142.3 ppm.

#### 3.1.2. (+/−)-*cis*-2-azido-1-indanol (**2**)

NaN_3_ (0.16 g, 2.5 mmol) was added to a stirred solution of **1** (0.5 g, 2.35 mmol) in anhydrous DMF (2.5 mL). The reaction mixture was stirred during 6 h at 80 °C and then diluted with water (8.0 mL). The aqueous solution was extracted with diethyl ether (3 × 10 mL). The organic phases were combined and dried over Na_2_SO_4_, filtrated, and the solvent was removed under reduced pressure until dryness. The residue was purified by p-TLC (*n*-hexane/diethyl ether; 1:1), affording compound **2** (0.330 g, 80%) as a white solid; m.p.: 130–132 °C; mp lit.: 129–131 °C [23]. ^1^H NMR (600 MHz, CDCl_3_) δ 2.40 (bs., 1H, CHOH), 3.17 (m, 2H, CH_2_), 4.36 (m, 1H, CHN_3_), 5.17 (d, *J* = 4.6 Hz, 1H, CHOH), 7.26–7.35 (m, 3H, ArH), 7.46 (d, *J* = 6.8 Hz, ArH-7); ^13^C NMR (150 MHz, CDCl_3_) δ 35.2 (CH_2_), 65.7 (CHN_3_), 76.4 (CHOH), 124.7, 125.1, 127.6, 129.0, 139.1, 141.8. 

#### 3.1.3. (+/−)-*cis*-2-amino-1-indanol (**3**)

Palladium/activated charcoal (10% Pd/C; 11 mg) was added to a stirred solution of compound **2** (90.5 mg, 0.5 mmol) prepared from compound **1**, as previously described [23,24] in absolute ethanol (10 mL)). The mixture was hydrogenated in a Parr hydrogenation apparatus at 40 psi of hydrogen pressure for 3 h. When the reduction was completed (TLC), the reaction was stopped, the reaction mixture was filtered, and the solvent was removed under reduced pressure until it reached dryness. Compound **3** was obtained (77.6 mg, 99%) as a yellow solid; m.p.: 130–132 °C. ^1^H NMR (300 MHz, CDCl_3_) δ 2.76 (dd, *J* = 5.1 Hz, *J* = 15.9 Hz, 1H, CHH), 3.13 (dd, *J* = 6.7 Hz, *J* =15.9, 1H, CHH), 3.61–3.65 (m, 1H, CHNH), 4.83 (d, *J* = 5.4 Hz, 1H, CHOH), 7.20–7.26 (m, 3H, ArH), 7.39–7.42 (m, 1H, ArH-7) ppm.^13^C NMR (75.4 MHz, CDCl_3_) δ 39.2 (CH2), 54.1 (CHNH_2_), 75.3 (CHOH), 125.1, 125.3, 127.0, 128.4, 141.0, 143.4 ppm.

#### 3.1.4. (+/−)-2-(5-amino-6-chloropyrimidin-4-yl-amino)-2,3-dihydro-1H-inden-1-ol (**4**)

A mixture of compound **3** (200 mg, 1.34 mmol) and 5-amino-4,6-dichloropyrimidine (220 mg, 1.34 mmol) in dry triethylamine (1.5 mL) and *n*-butanol (5.3 mL) was heated at reflux for 24 h under argon atmosphere. Then, the reaction mixture was cooled and the solid obtained was filtered and washed with EtOAc (2 × 5 mL) to afford compound **4** as a grey solid (150 mg, 40%); m.p.: 172–174 °C. ^1^H NMR (600 MHz, DMSO-*d6*) δ 3.03 (dd, *J* = 8.5 Hz, *J* = 15.4 Hz, 1H, CHH), 3.12 (dd, *J* = 7.5 Hz, *J* = 15.4 Hz, 1H, CHH), 4.54–4.64 (m, 1H, CHNH), 5.02 (t, *J* = 5.3 Hz, 1H, CHOH), 5.18 (brs, 2H, NH_2_), 5.25 (d, *J* = 5.4 Hz, 1H, OH), 6.83 (d, *J* = 6.3 Hz, 1H, CHNH), 7.21–7.28 (m, 3 H, ArH), 7.39 (d, 1H, *J* = 7.2 Hz, ArH-7), 7.77 (s. 1H, N=CH–N) ppm; ^13^C NMR (150 MHz, DMSO-*d6*) δ 35.9 (CH_2_), 55.6 (CHNH), 72.4 (CHOH), 124.3, 125.1, 125.9 (C7), 127.1, 128.6, 137.4, 141.5, 144.4, 146.1 (N=CH–N), 152.5 ppm. HRMS-ESI *m*/*z* [M + H]^+^ calcd for C_13_H_14_N_4_ClO: 277.0856, found: 277.0859.

#### 3.1.5. (+/−)-2-(6-chloro-9H-purin-9-yl)-2,3-dihydro-1H-inden-1-ol (**5**) 

A mixture of compound **4** (80 mg, 0.29 mmol), triethylorthoformate (1.8 mL), and HCl 12 N (0.1 mL) was stirred at room temperature under argon atmosphere for 2 h. Then, the solvent was removed under reduced pressure until it reached dryness, and the product was purified by p-TLC (eluant EtOAc:Hexane 1:1) to afford compound **5** as a solid (60 mg, 73%); m.p.: >220 °C with decomposition. ^1^H NMR (600MHz, DMSO-*d6*) δ 3.46 (dd, *J* = 7.8 Hz, *J* = 16.3 Hz, 1H, CHH), 3.73 (dd, *J* = 8.4 Hz, 15.8 Hz, 1H, CHH), 5.14 (t, *J* = 5.7 Hz, 1H, CHOH), 5.40–5.44 (m, 1H, CHN), 5.54 (bs, 1H, OH), 7.29–7.40 (m, 3H, ArH), 7.43 (d, *J* = 7.4 Hz, 1H, ArH-7), 8.60 (s, 1H, (N=CH–N), 8.81 (s, 1H, (N=CH–N) ppm, ^13^C NMR (150 MHz, DMSO-*d6*) δ 35.1 (CH_2_), 58.2 (CHN), 73.1 (CHOH), 125.2, 126.0, 127.7, 129.3, 140.5, 143.0, 147.4, 149.2, 151.9, 153.1 ppm. HRMS-ESI *m*/*z* [M + H]^+^ calcd for C_14_H_12_ClN_4_O: 287.0700, found: 287.0695.

#### 3.1.6. 3-((+/−)1-hydroxy-2,3-dihydro-1*H*-inden-2-yl)-3*H*-[1,2,3]triazolo[4,5-d]pyrimidin-7-ol (**7**)

A cooled solution (0 °C) of compound **4** (100 mg, 0.36 mmol) in HCl 1N (1.3 mL) was treated with a solution of sodium nitrite (36 mg, 0.48 mmol) in water (3.3 mL). The mixture was stirred and allowed to warm up to room temperature, and then heated at reflux for 1 h. Finally, the solvent was removed under reduced pressure until it reached dryness. The solid residue was purified by p-TLC (eluant EtOAc) to afford compound **7** as a white solid (70 mg, 72%), m.p.: 178–180 °C. ^1^H NMR (600 MHz, DMSO-*d6*) δ 3.49 (dd, *J* = 7.8 Hz, *J* = 16.0 Hz, 1H, CHH), 3.98 (dd, *J* = 6.3 Hz, *J* = 16.0 Hz, 1H, CHH), 5.34 (t, *J* = 6.2 Hz, 1H, CHOH), 5.39 (d, *J* = 6.3 Hz, 1H, OH), 5.45–5.55 (m, 1H, CHN), 7.26–7.41 (m, 4H, ArH), 8.26 (s, 1H, N=CH–N), 12.66 (s, 1H, N=C–OH) ppm; ^13^C NMR (150 MHz, DMSO-*d6*) δ 34.1 (CH_2_), 61.4 (CHN), 74.3 (CHOH), 125.0, 125.6, 127.4, 128.9, 129.9, 140.6, 143.0, 149.6, 149.9, 155.9 ppm. HRMS-ESI *m*/*z* [M + Na]^+^ calcd for C_13_H_11_N_5_NaO_2_: 292.0810, found: 292.0807.

#### 3.1.7. (+/−)2-(7-amino-3*H*-[1,2,3]triazolo[4,5–d]pyrimidin-3-yl)-2,3-dihydro-1*H*-inden-1-ol (**8**)

A solution of sodium nitrite (33 mg, 0.48 mmol) in water (3.7 mL) was added to a cooled (0 °C) solution of compound **4** (100 mg, 0.36 mmol) and 1N HCl (1 mL) The mixture was stirred at 0 °C for 15 min, and then NH_4_OH (2 mL) was added and the mixture was heated under reflux for 5 min. The solvent was removed under reduced pressure, and the solid residue was purified by p-TLC (eluant EtOAc) to afford compound **8** as a white solid (80 mg, 82%); m.p.: >250 °C with decomposition. ^1^H NMR (600 MHz, DMSO-*d6*) δ 3.49 (dd, *J* = 8.0 Hz, *J* = 16.0 Hz, 1H, CHH), 4.05 (dd, *J* = 6.7 Hz, *J* = 16.0 Hz, 1H, CHH), 5.30–5.37 (m, 2H, CHOH and OH), 5.35–5.60 (m, 1H, CHN), 7.26–7.42 (m, 4H, ArH), 8.00 (bs, 1H, N=CH–NH), 8.30 (s, 1H, N=CH–NH), 8.34 (bs, 1H, C=NH) ppm; ^13^C NMR (150 MHz, DMSO-*d6*), δ 34.0 (CH_2_), 61.0, (CHN), 74.1, (CHOH), 124.3, 125.0, 125.7, 127.3 128.9, 140.8, 143.2, 150.1, 156.6, 156.8 ppm. HRMS-ESI *m*/*z* [M + Na]^+^ calcd for C_13_H_12_N_6_NaO_2_: 291.0973, found: 291.0965.

#### 3.1.8. (+/−)-2-(4-hydroxymethyl-1*H*-1,2,3-triazol-1-yl)-2,3-dihydro-1*H*-inden-1-ol (**9**)

Propargyl alcohol (66 μL, 1.14 mmol) and compound **2** (100 mg, 0.57 mmol) were suspended in 2 mL of a water/tert-butanol mixture (1:1). Sodium ascorbate (0.3 mmol, 300 μL of freshly prepared 1 M solution in water) was added, followed by copper (II) sulfate pentahydrate (7.5 mg, 0.03 mmol, in 100 μL of water). The heterogeneous mixture was stirred vigorously overnight, at which point it cleared and the TLC analysis indicated total consumption of the reactants. The reaction mixture was diluted with 10 mL of water and the brown precipitate was collected by filtration. After being washed with cold water (2 × 5 mL), the precipitate was dried under vacuum to afford compound **9** as a brown solid (70 mg, 53%); m.p.: 131–133 °C. ^1^H NMR (600 MHz, DMSO-*d6*) δ 3.40 (dd, *J* = 7.5 Hz, *J* = 15.9 Hz, 1H, CHH), 3.45 (dd, *J* = 6.7 Hz, J = 15.4 Hz, 1H, CHH), 4.50 (m, *J* = 5.42 Hz, 2H CH_2_OH), 5.14 (t, *J* = 5.7 Hz, 1H, CH_2_OH), 5.18 (bs, *J* = 5.6 Hz, 1H, CHOH), 5.37 (ddd, *J* = 5.6Hz, 6.7 Hz, 7.5 Hz, 1H, CHN), 5.58 (bs, 1H, CHOH), 7.28–7.36 (m, 3H, ArH), 7.40 (d, *J* = 7.1 Hz, 1H, ArH), 7.68 (s, 1H, N–CH=C); ^13^C NMR (150 MHz, DMSO-*d6*) δ 36.0 (CH_2_), 55.6 (CH_2_OH) 63.8, (HCN), 74.2, (HCOH), 122.7 (NCH=C), 125.1, 125.5, 127.6 129.0, 140.2, 143.3, 147.8 (N–C=C). HRMS-ESI *m*/*z* [M + H]^+^ calcd for C_12_H_14_N_3_O_2_: 232.1086, found: 232.0988.

### 3.2. Evaluation of Biological Activity

#### 3.2.1. Cells

The HCV genotype 1b replicon cell line, Con1/SG-Neo (I)—Huh7.5, cells containing the subgenomic (SG) HCV replicon (Huh 7.5 SG, Catalogue number APC50) was donated by APATH (New York, NY, USA). These cells were maintained at 37 °C in a humidified atmosphere of 5% CO in a subconfluent state in DMEM (Life Technologies, Carlsbad, CA, USA) supplemented with 10% FBS (Life Technologies Corp., Carlsbad, CA, USA) and 750 µg/mL G418 (Life Technologies, Carlsbad, CA, USA).

#### 3.2.2. Drug Preparation

Compounds **3**, **5**, **7**, **8**, **9**, 6-chloropurine, and morphine (Sigma-Aldrich, St. Louis, MO, USA) were solubilized in DMSO. Naloxone (Denver Farma S.A., Buenos Aires, Argentina) was solubilized in physiological solution.

Unless described in detail elsewhere, the different compounds and drugs were used in the following final concentrations: (i) **3**, **5**, and 6-chloropurine: 25.00 μM; (ii) **7**–**9**: 100.00 μM; (iii) morphine: 1.00 μM; and (iv) naloxone 0.01 μM.

The drug concentrations used for the assays were defined according to: (i) The cytotoxicity effect exerted on the Huh 7.5 SG cell line for the four newly synthesized carbanucleosides (compounds **5**, **7**–**9**); (ii) the equivalent amount of the precursor on the final structure of carbanucleoside **5** for 6-chloropurine and compound **3**; and (iii) the previous results reported by Li et al. [26] for morphine and naloxone.

#### 3.2.3. Cytotoxic Assays

The Huh7.5 SG cells were seeded in 96-well plates (2.0 × 10^4^ cells/well), grown overnight and treated with the carbanucleoside analogues synthesized at different concentrations (1.56 to 100.00 µM). The cytotoxic activity was determined after 96 h of incubation for the HCV replicon cells using the 3-(4,5-dimethylthiazol-2-yl)-5-(3-carboxymethoxyphenyl)-2-(4-sulfophenyl)-2*H*-tetrazolium/phenazine methosulfate (MTS/ PMS) assay Promega, Madison, WI, USA). Absorbance at λ = 495 nm was determined using FlexStation 3 (Molecular Devices, San Jose, CA, USA).

#### 3.2.4. HCVg1b Replication Assay

The Huh7.5 SG cells were incubated for 96 h in 6-well plates (5.0 × 10^5^ cells/well) in the presence of different drugs, alone or combined. Samples were collected every 24 h and the medium of each well was completely renewed using fresh medium supplemented with 5% FBS without G418 (to eliminate the potential loss of cells due to the reduction of the HCV replicon copy number and G418 resistance). HCVg1b replication was determined by the quantification of viral RNA at 48 and 72 h post-incubation in the supernatants of cell cultures by RT-qPCR using the COBAS^®^ AmpliPrep/COBAS^®^ TaqMan^®^ HCV Quantitative Test (Roche, Basel, Switzerland). Untreated cultures were included in each assay as negative controls (100% of HCV viral load). Viral RNA levels were normalized to each untreated control (100% of HCV viral load).

### 3.3. Molecular Modelling and Superimpose Structures

Morphine and both enantiomers of compound **5** were optimized with the density functional theory (DFT) using the B3LYP functional and 6-31G* basis set. The quantum calculations included full optimization of the geometric structures and the scan of the dihedral angle, which connects both bicyclic rings of compound **5**. Then, the 10 conformational minima for each enantiomer were superimposed on the morphine pharmacophore group. For this purpose, the Chimera v1.12 software (Resource for Biocomputing, Visualization, and Informatics—University of California, San Francisco, CA, USA) was used for obtaining the root mean square deviation (RMSD) of the 10 superimposed atoms. The best RMSD was obtained for the conformer of the second minimum potential energy for each enantiomer.

## 4. Conclusions

We report here a convenient synthetic procedure for the preparation of novel three purinyl- and 8-azapurinyl-carbanucleoside derivatives, obtained as a racemic mixture from *cis*-2-amino-1-indanol, as well as one triazolyl analogue obtained from racemic *cis*-2-azido-1-indanol. The action of these compounds against hepatitis C virus was evaluated in vitro using the replicon cell line, Huh7.5 SG. All the nucleoside analogues studied here promoted viral replication. Several biological experiments and computational studies were performed to elucidate the effect of these compounds on HCVg1b replication. Based on the results of all the experiments performed, we propose that the increase in HCVg1b replication could be mediated, at least in part, by a similar mechanism to that of morphine on the enhancement of this replication. The presence of opioid receptors in the Huh7.5 SG cells was indirectly determined in this work for the first time.

The findings of this work impact on the design of nucleoside analogues because, surprisingly, the compounds of a family that is commonly known for its antiviral effect could be inactive ones. Nevertheless, it is unlikely that they show the opposite effect as reported in this work. Our results provide relevant molecular information to gain further knowledge about this system. Moreover, these open the possibility of considering other compounds that enhance viral replication in the recurrence of HCV infection, such as morphine in addiction cases.

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
