# Peer review of "Synthesis of New Indanyl Nucleoside Analogues and their Biological Evaluation on Hepatitis C Virus (HCV) Replicon"

_molecules, 2019, doi:10.3390/molecules24050990_

Round 1

Reviewer 1 Report

In the submitted manuscript, Gomez et al. report the "Synthesis of New Indanyl Nucleoside Analogues and their Biological Evaluation on Hepatitis C Virus 3 (HCV) Replicon".

The authors report a convenient access (few steps and good overall yields) to indanyl nucleoside analogues as a racemic mixture starting from indene. Only four analogues were described, while a series of compounds could be easily prepared and tested following this scheme. In particular conversion of 5 into the corresponding hypoxanthine and adenine derivatives was not further exploited. It had been appropriate to measure toxicity and viral activity of these compounds in comparison with 5, 7,8.

The authors found that these molecules did not inhibit but rather promoted the HCVg2 replication in vitro (RNA levels). They showed a time-dependent effect with 3 and 5, but data with compounds 7-9 are missing in this Fig. 4. The authors next investigate the possible relationship between these nucleoside analogues and opioid compounds. They validated the effect of morphine and naloxone (i.e. presence of opioid receptors) on viral replication in Huh7.5 cells. Using morphine, naltrexone (a pan-opioid receptor antagonist) and compound 5, intermediate levels in RNA viral load were observed (partial decrease in both case). There are no conclusive results. As the affinity of morphine for opioid receptor is high, the relative concentrations of morphine and 5 could be important. There are absent in Fig 8 and experimental section. Data with the most "active" compounds 7-9 are also missing. The last conclusion based on the surperimposition of morphin and 5 (each eniantiomer) structures is speculative in my opinion.

The authors present a convenient synthesis of new indanyl nucleoside, but only three derivatives were isolated. Experimental procedures are well detailled, the final (tested) compounds fully characterized (NMR and HRMS). In an effort to explain the unexpected enhancement of viral load in the presence of these analogues, the authors suggested a mode of action mediated by opioid receptor, but with no conclusive results. How these results may impact the design of nucleoside analogues ?

The major question is - Are these results important and relevant for publication in Molecules ?

Minor comments :

lane 102 : Legend should be changed.    Effect on vialability (and not on viral load)

lane 171.  Legend Fig 8 Concentration used for morphine and 5 should be added.

lane 409 : p4540-4552

Author Response

The authors report a convenient access (few steps and good overall yields) to indanyl nucleoside analogues as a racemic mixture starting from indene. Only four analogues were described, while a series of compounds could be easily prepared and tested following this scheme. In particular conversion of 5 into the corresponding hypoxanthine and adenine derivatives was not further exploited. It had been appropriate to measure toxicity and viral activity of these compounds in comparison with 5, 7,8.

As you point out, it would have been convenient to make the measurements that you suggest. However, the derivatives that you propose could not be prepared from 5, as this intermediate is decomposed in the attempt to transform it into the hydroxy or amine derivative.

The authors found that these molecules did not inhibit but rather promoted the HCVg1b replication in vitro (RNA levels). They showed a time-dependent effect with 3 and 5, but data with compounds 7-9 are missing in this Fig. 4. 

As we found that the compounds promoted viral replication, we considered it unnecessary to study this effect as a function of time. The aim of our work then became the search for the atomic grouping present in the molecule that is responsible for such unexpected effect. The bibliographic revision related to the subject in question led us, to the extent possible, to the work conducted by Li et al. on morphine, as a compound that increases HCV viral load. Li et al. found that the morphine effect drops over time and that is the reason why we decided to do the reading at 48 h.   Particularly, we only conducted the studies using compound 5, because it allowed us to use the corresponding molecular fragments to try to establish the atomic grouping responsible for the effect observed.   

The authors next investigate the possible relationship between these nucleoside analogues and opioid compounds. They validated the effect of morphine and naloxone (i.e. presence of opioid receptors) on viral replication in Huh7.5 cells. Using morphine, naltrexone (a pan-opioid receptor antagonist) and compound 5, intermediate levels in RNA viral load were observed (partial decrease in both case). There are no conclusive results. As the affinity of morphine for opioid receptor is high, the relative concentrations of morphine and 5 could be important. There are absent in Fig 8 and experimental section. Data with the most "active" compounds 7-9 are also missing

As we have already mentioned, the activity of these derivatives has no potential pharmacological application and since cell system availability is limited in our laboratories, we decided to design an experiment that would allow us to advance in the structure-activity relationship of the derivative, which could be compared with its molecular fragments separately. In relation to Fig.8, we have added the corresponding concentrations requested by the reviewer. In the experimental section, item 4.2.2 has had the used concentration of all the compounds assayed since the original manuscript.

The last conclusion based on the superimposition of morphine and 5 (each enantiomer) structures is speculative in my opinion.

In reference to the opinion of the reviewer, we do not agree, given the following findings:

1) The increase in viral load in the presence of compound 5, similar to that described by Li et al against morphine,

2) The decrease in the viral load, when the cellular system was previously incubated with a specific opioid receptor antagonist.

From these experimental results, we strongly believe that the RMSD values determined for the superimposition studies of the pharmacophore of morphine with our compounds, are relevant and valid.  

The authors present a convenient synthesis of new indanyl nucleoside, but only three derivatives were isolated. Experimental procedures are well detailled, the final (tested) compounds fully characterized (NMR and HRMS). In an effort to explain the unexpected enhancement of viral load in the presence of these analogues, the authors suggested a mode of action mediated by opioid receptor, but with no conclusive results. How these results may impact the design of nucleoside analogues?

We think that our work has great impact on the design of nucleoside analogues because, surprisingly, although these type of compounds are commonly known for their antiviral effect, new derivatives could be inactive, but it is unlikely that they could have the opposite effect.  Perhaps other researchers have found similar results to ours in nucleoside chemistry but have not reported them.  Our findings provide relevant molecular information for the design of nucleoside analogues as potential drugs. (MIO)

The major question is - Are these results important and relevant for publication in Molecules?

We consider that the research involved herein is appropriate and relevant for this journal as it let us shed light, from molecular structure, the unexpected effect of a group of nucleoside analogues that were prepared with the aim of being potentially effective in the treatment of HCV infection.

Minor comments :

lane 102 : Legend should be changed.    Effect on vialability (and not on viral load) DONE

lane 171.  Legend Fig 8 Concentration used for morphine and 5 should be added. DONE

lane 409 : p4540-4552 DONE

Reviewer 2 Report

Synthesis of New Indanyl Nucleoside Analogues and 2 their Biological Evaluation on Hepatitis C Virus 3 (HCV) Replicon

Gomez et al. investigate the effect of four indanyl carbanucleoside on  hepatitis C virus infection in vitro, using the model of replicon cell  line Huh7.5 SG as recipient cells. The experimental approach to demonstrate that the compounds are not efficacious as hepatitis C infection inhibitors is weak since the authors have shown that the experiments were performed in replicates only after 24 and not 48 and 72 hours. Anyhow it is likely that the compounds were not active. Then the authors shift their interest toward a complete different use of the compounds on the basis of pharmacophore investigation. The experimental approach is weak since does not seem that the experiments were performed in replicates. In addition their conclusion is superficial since they do not show data on the mechanisms in particular  if  the effect  of the compounds is opioid receptor mediated or  other. In addition the experiments of combined treatment with  compound 5 and Morphine are not properly described and the conclusions are not clearly illustrated. I think that this section of the paper would deserve a dedicated manuscript with more detailed investigations regarding  possible mechanisms involved.

The English form needs to be revised by a native language expert. Collectively also on the basis of description of the experimental work , the entire manuscript is not  very understandable.

In conclusion this paper is rather disorganized and it has to be rejected

Author Response

Gomez et al. investigate the effect of four indanyl carbanucleoside on hepatitis C virus infection in vitro, using the model of replicon cell line Huh7.5 SG as recipient cells. The experimental approach to demonstrate that the compounds are not efficacious as hepatitis C infection inhibitors is weak since the authors have shown that the experiments were performed in replicates only after 24 and not 48 and 72 hours. Anyhow it is likely that the compounds were not active. Then the authors shift their interest toward a complete different use of the compounds on the basis of pharmacophore investigation.

We guess that the reviewer would not have fully understood the protocol used. In Materials & methods it is indicated that the cells were maintained with the compounds under study until 96hs, both to evaluate the cytotoxicity and to determine viral loads. These were read at 48 hours and/or at 72 hours of incubation as indicated in the manuscript and in all cases an increase in viral load was observed.

The experimental approach is weak since does not seem that the experiments were performed in replicates.

Considering that the cellular system used was donated in a limited way by APATH (Brooklyn, NY, USA) for these investigations, and that the determinations of RT-qPCR have a significant cost in the external services to our lab, all the experimental points were performed in duplicate, with consistency in the tendency obtained between them.

In addition their conclusion is superficial since they do not show data on the mechanisms in particular if the effect of the compounds is opioid receptor mediated or other. In addition the experiments of combined treatment with compound 5 and Morphine are not properly described and the conclusions are not clearly illustrated. I think that this section of the paper would deserve a dedicated manuscript with more detailed investigations regarding possible mechanisms involved.

Perhaps, the reviewer misinterpreted the purpose of the manuscript. The unexpected results obtained gave rise to other investigations.  So far we consider it important to have established in an indirect way the presence of opioid receptor in this cellular system and to explain that, at least partially, the effect of the new compounds can occur through it. In a future work, we will try to establish the mechanism involved, as you suggest. The experimental details requested by the reviewer can be found in items 4.2.2 and 4.2.4, as well as in the analysis of figures 7 and 8.

The English form needs to be revised by a native language expert. Collectively also on the basis of description of the experimental work, the entire manuscript is not very understandable.

DONE

Reviewer 3 Report

In this paper the authors describe the procedure for preparation of new indanyl derivatives, triazolyl, purinyl and 8-azapurinyl, obtained as a racemic mixture from cis-2-azido-1-indanol. All new compounds were well characterized using spectroscopic methods. Biological experiments and computational studies were performed to elucidate the effect of these compounds on HCVg2 replication. The authors concluded that the presence of opioid receptors in Huh7.5 SG cells mediated by naloxone was indirectly determined. The work is well presented and the results are of considerable interest. However, I would suggest a native English speaker to check the manuscript throughout as multiple language errors are still present.

I recommend that presented manuscript can be accepted for publication regarding chemical synthesis part providing that all issues raised are appropriately dealt with:

Line 18: Consider that one of the new compounds is triazolyl derivate of indane.

Line 19-23: Please consider rewriting of the Abstract. The most important results and conclusions would be written in abstract. It would be appropriate to do analysis and comments in the section Results and Discussion.

Line 66: In reference [21] was described the synthesis of bromohydrin by reduction of bromoketone. For the synthesis of bromohydrin from the indene it would be appropriate to add the following reference: Adv. Synth. Catal. 2005, 347, 255;

Line 70-74: This section appears confusing. For this discussion provide appropriate NMR spectra in supporting information.

Line 76: It would be appropriate to add that indanyl carbonucleosides were prepared using the classical approach to carbocyclic nucleosides of nucleosides. (Reference [20] Nucleosides Nucleotides 1998, 17, 1237-1253);

Line 80-81: For this discussion provide appropriate NMR spectra in supporting information.

Line 85: Triazole 9 was prepared by well-known click reaction. It would be appropriate to add discussion and reference about well-known click reaction (suggestion: ACIEE 2001, 40, 2004-2021).

Line 117: 5 should be bold;

Line 146 and line 150: structure of Meperidine (CAS 57-42-1) is not correct. It would be appropriate to remove discussion about meperidine.

Line 193: It would be considered that triazolyl derivate 9 was not obtained from cis-2-amino-1-indanol!

Line 211, 233 and 300: Complex absorption (ca) is not applicable for description of signals in NMR spectra. It would be appropriate to use multiplet (m) for complex NMR signals.

Line 232: Literature melting point for compounds 2 was not described in reference [25]. It would be appropriate to add reference for this information [22] Tetrahedron: Asymmetry 2005, 16, 3633–3639;

Line 387: In reference [12] instead volume 42 write volume 442;

Line 409: In reference [21] instead pages 4550-4552 write pages 4540-4542.

Author Response

In this paper the authors describe the procedure for preparation of new indanyl derivatives, triazolyl, purinyl and 8-azapurinyl, obtained as a racemic mixture from cis-2-azido-1-indanol. All new compounds were well characterized using spectroscopic methods. Biological experiments and computational studies were performed to elucidate the effect of these compounds on HCVg1b replication. The authors concluded that the presence of opioid receptors in Huh7.5 SG cells mediated by naloxone was indirectly determined. The work is well presented and the results are of considerable interest. However, I would suggest a native English speaker to check the manuscript throughout as multiple language errors are still present.

DONE

I recommend that presented manuscript can be accepted for publication regarding chemical synthesis part providing that all issues raised are appropriately dealt with:

Line 18: Consider that one of the new compounds is triazolyl derivate of indane.

The reviewer is right. However, some drugs like Rivabirin has a triazolyl ring as a “nucleobase” and are considered a carbonucleoside analogue, so we consider that compound 9 could be included as an indanyl carbanucleoside derivative.

Line 19-23: Please consider rewriting of the Abstract. The most important results and conclusions would be written in abstract. It would be appropriate to do analysis and comments in the section Results and Discussion.

 We re-elaborate the abstract having into account these suggestions.

Line 66: In reference [21] was described the synthesis of bromohydrin by reduction of bromoketone. For the synthesis of bromohydrin from the indene it would be appropriate to add the following reference: Adv. Synth. Catal. 2005347, 255;

DONE

Line 70-74: This section appears confusing. For this discussion provide appropriate NMR spectra in supporting information.

We have included the spectrum in supplementary material section.

Line 76: It would be appropriate to add that indanyl carbonucleosides were prepared using the classical approach to carbocyclic nucleosides of nucleosides. (Reference [20] Nucleosides Nucleotides 1998, 17, 1237-1253);

DONE

Line 80-81: For this discussion provide appropriate NMR spectra in supporting information.

We have included the spectrum in supplementary material section.

Line 85: Triazole 9 was prepared by well-known click reaction. It would be appropriate to add discussion and reference about well-known click reaction (suggestion: ACIEE 2001, 40, 2004-2021).

DONE

Line 117: 5 should be bold;

DONE

Line 146 and line 150: structure of Meperidine (CAS 57-42-1) is not correct. It would be appropriate to remove discussion about meperidine.

We have corrected this part of manuscript including the correct compound that we are talking about: prodine.

Line 193: It would be considered that triazolyl derivate 9 was not obtained from cis-2-amino-1-indanol.

It was corrected.

Line 211, 233 and 300: Complex absorption (ca) is not applicable for description of signals in NMR spectra. It would be appropriate to use multiplet (m) for complex NMR signals.

DONE

Line 232: Literature melting point for compounds 2 was not described in reference [25]. It would be appropriate to add reference for this information [22] Tetrahedron: Asymmetry 200516, 3633–3639;

DONE

Line 387: In reference [12] instead volume 42 write volume 442;

DONE

Line 409: In reference [21] instead pages 4550-4552 write pages 4540-4542.

DONE

Round 2

Reviewer 2 Report

The version has been improved. In this form can be accepte